# Changes in Visual Function and Correlations with Inner Retinal Structure in Acute and Chronic Leber’s Hereditary Optic Neuropathy Patients after Treatment with Idebenone

**DOI:** 10.3390/jcm10010151

**Published:** 2021-01-04

**Authors:** Berthold Pemp, Christoph Mitsch, Karl Kircher, Andreas Reitner

**Affiliations:** Department of Ophthalmology, Medical University of Vienna, 1090 Vienna, Austria; christoph.mitsch@meduniwien.ac.at (C.M.); karl.kircher@meduniwien.ac.at (K.K.); andreas.reitner@meduniwien.ac.at (A.R.)

**Keywords:** Leber’s hereditary optic neuropathy (LHON), idebenone, optic nerve, vision, medical treatment

## Abstract

Progressive impairment and degeneration of retinal ganglion cells (RGC) and nerve fibers in Leber’s hereditary optic neuropathy (LHON) usually cause permanent visual loss. Idebenone is currently the only approved treatment. However, its therapeutic potential in different stages of LHON has not been definitely clarified. We aimed to investigate the changes in visual function and correlations with retinal structure in acute and in chronic LHON patients after treatment with idebenone. Twenty-three genetically confirmed LHON patients were followed during treatment using logMAR charts, automated perimetry and optical coherence tomography (OCT). Mean visual acuity improved significantly in acute patients treated within 1 year from onset (−0.52 ± 0.46 logMAR from nadir), in early chronic patients who started after 1–5 years (−0.39 ± 0.27 logMAR from baseline), and in late chronic patients with treatment initiation after >5 years (−0.33 ± 0.28 logMAR from baseline, *p* < 0.001 all groups). In acute and in chronic patients, strong correlations between OCT and visual function parameters were present only after treatment. This and the sustained visual recovery after treatment may indicate a reactivated signal transduction in dysfunctional RGC that survive the acute phase. Our results support previous evidence that idebenone has therapeutic potential in promoting visual recovery in LHON.

## 1. Introduction

Leber’s hereditary optic neuropathy (LHON) is a rare inherited disease induced by mitochondrial dysfunction, which is typically caused by point mutations in genes of the mitochondrial DNA. It has an estimated prevalence of 1 in 45,000 people in western countries and predominantly affects young adult males [1]. The disease is characterized by progressive impairment of macular retinal ganglion cells (RGC) resulting in acute or subacute painless and severe reduction of visual acuity (VA) in both eyes. The acute phase subsequently leads to degeneration of a large amount of RGC and retinal nerve fibers, which results in optic atrophy and permanently reduced VA in most cases [2]. Most patients experience bilateral vision impairment to 20/200 or less during the first year after disease onset, nearly all patients have permanent vision loss due to central scotoma and most of them are legally registered as visually impaired [3].

Three different mutations in the mitochondrial DNA (m.11778G>A, m.3460G>A and m.14484T>C) in genes encoding for different components of complex I of the mitochondrial respiratory chain have been identified as genetic causes in about 95% of patients [2]. Their impact on mitochondrial oxidative phosphorylation (OXPHOS) has been studied intensively over the last decades. Common features of cells bearing LHON mutations include reduced mitochondrial respiration rate, augmented reactive oxygen species (ROS) production, and also increased mitophagy [4,5]. The well-established pathogenetic characterization enabled the development of targeted pharmaceutical treatments including different antioxidants [6]. Gene therapy for allotopic expression of the wildtype ND4 protein is also currently tested as a treatment in LHON patients, but limited to the most frequent mutation m.11778G>A [6].

The short-chain benzoquinone idebenone has been approved by the European Medicine Agency for the treatment for LHON under the status of an orphan drug and currently is the only available drug therapy for LHON, regardless of the causing mutation. It acts as a potent radical scavenger [7], supports electron transfer in the mitochondrial respiratory chain [8] and reduces mitophagy [5]. The beneficial use of idebenone in LHON has been supported by results from two controlled studies [9,10] and from a subsequent expanded access program [11], which led to its approval for this hitherto untreatable mitochondrial disease. Due to limited evidence in longer-lasting LHON, an international consensus statement recommended idebenone as standard therapy for genetically confirmed LHON only in the first year after disease onset [12]. However, recently we were able to demonstrate a significant improvement in visual function also in patients with chronic LHON, when treatment with idebenone was initiated later than 5 years after disease onset [13].

The course of visual function and retinal tissue loss in LHON over time has been described in various studies [14,15,16,17]. However, structure-function relationships in patients treated with idebenone have not yet been evaluated. Hence, this exploratory study aims to investigate the changes in VA and their correlation with inner retinal layer morphology in acute and chronic LHON patients during treatment with idebenone.

## 2. Methods

Data of all genetically confirmed LHON patients treated at our department with a minimum follow-up of 12 months after treatment start were included in this retrospective study. All patients had visual symptoms in both eyes due to LHON before treatment start and none of the patients had any additional neurologic symptoms or cardiac disorders. All patients received oral idebenone (Raxone^®^, Santhera Pharmaceuticals, Pratteln, Switzerland) at the approved dose of 900 mg per day. Following current recommendations [12], they were examined every three months during treatment and in broader intervals after treatment discontinuation. Treatment was discontinued if VA and visual fields were stable over a period of 12 months without further significant improvement. Best corrected VA was tested using illuminated logarithmic reading charts (ETDRS charts, Precision Vision, La Salle, CA, USA) taking into account all letters correctly identified on the chart, and was notated as logarithm of the minimum angle of resolution (logMAR). A change of at least −0.2 logMAR corresponding to an improvement of 10 letters on the ETDRS chart or a change from off-chart to on-chart was defined as clinically relevant improvement (CRI) of VA. A change of this size is commonly accepted as being clinically relevant in LHON and represents a significant change in vision related quality of life in optic neuropathy [18]. Visual field defects were quantified by measurement of mean deviation (MD) in the central 30 degree visual field using automated static threshold perimetry (HFAII 30-2 Threshold Test, Carl Zeiss Meditec, Jena, Germany). To allow for structure-function analysis in the macula, the six points nasal to the blind spot in the 30 degree test grid were defined as the central cluster of visual field, and central deviation (CD) was calculated as the average from the numerical total deviation map. At each visit macular ganglion cell layer (GCL) volume and peripapillary retinal nerve fiber layer (pRNFL) thickness were measured with spectral domain optical coherence tomography (Spectralis OCT, NSite module, software version 6.9a, Heidelberg Engineering, Heidelberg, Germany) using a 20 × 20 degree volume scan centered on the macula, and a 12 degree ring scan centered on the optic nerve head, respectively. Scans at baseline were defined as reference location for all follow-up scans. The built-in segmentation software was used for automated layer segmentation. GCL volume was defined as the layer volume within the central 6 mm diameter of the macula.

Approval by the ethics committee of the Medical University of Vienna was obtained and examination data were retrospectively analyzed. Since visual function in LHON is known to develop differently during the successive stages, patients were divided into three groups according to their time from LHON onset until treatment initiation including eight acute patients (<1 year, 16 eyes), seven early chronic patients (1 to 5 years, 14 eyes) and eight late chronic patients (>5 years, 16 eyes). The statistical analysis was conducted using the software STATISTICA (Release 6.1; StatSoft Inc., Tulsa, OK, USA). Numerical data were expressed as means ± standard deviation (SD). Age, duration of observation and treatment, VA and OCT data were approximately normally distributed in all groups, as assessed by the Shapiro-Wilk test (*p* > 0.05). Changes in VA and OCT were evaluated using ANOVA for repeated measurements. MD was not normally distributed as assessed by the Shapiro-Wilk test (*p* < 0.05). Changes in visual field parameters were evaluated using the Wilcoxon signed-rank test. Pearson’s correlation coefficients were calculated to evaluate associations between changes in visual function during treatment and biometric measurements at baseline, and to evaluate correlations between OCT and visual function parameters at baseline and at the last available visit. A *p*-value of 0.05 was considered the level of significance for all calculations. The *p*-values were not corrected for testing multiple outcomes due to the exploratory character of the study.

## 3. Results

Patient characteristics of all 3 groups are presented in Table 1. Age, observation time from treatment start and mean duration of treatment were comparable between the groups (*p* > 0.15, unpaired *t*-test).

### 3.1. Changes in Visual Acuity

VA at baseline was similar between groups, but eight eyes of five acute patients showed a marked decline in the first year reaching a minimum after 3 to 9 months. During treatment, a significant increase in mean VA was observed in acute patients and in both chronic groups over time (Figure 1; acute: *p* = 0.01, early chronic: *p* < 0.00001, late chronic: *p* = 0.00001). The improvement was statistically significant in the acute group at month 12, and in both chronic groups already at month 3. Maximum VA gain was statistically significant in all groups (Figure 2; acute: −0.52 ± 0.46 logMAR from nadir, *p* = 0.0004; early chronic: −0.39 ± 0.27 logMAR from baseline, *p* = 0.0001; late chronic: −0.33 ± 0.28 logMAR from baseline, *p* = 0.0003). The percentage of responding eyes with CRI was 56%, 71% and 69% in acute, early chronic and late chronic patients, respectively. VA gain in eyes with CRI was higher in acute patients (Table 1; acute: −0.82 ± 0.39 logMAR from nadir; early chronic: −0.48 ± 0.26 logMAR; late chronic: −0.43 ± 0.29 logMAR; *p* < 0.037 between acute and chronic groups).

CRI in at least one eye was reached in all mutations except for the patients with the m.14495A>G or m.13051G>A mutations, who had only mild visual disturbance at treatment start. In the acute group, all eyes of the three patients with the m.14484T>C mutation showed a marked improvement during treatment with normalized VA (≤0.0 logMAR) at the last visit in two patients. The other two acute patients with CRI had the mutations m.11778G>A and m.13513G>A, respectively. Among the early chronic cases, both m.11778G>A patients and three m.3460G>A patients showed a CRI. Three early chronic patients reached a final VA ≤ 0.3 logMAR in both eyes. All late chronic patients with the three common LHON mutations had a CRI in at least one eye, but VA increase in patients with the mutation m.14484T>C was not higher than in the m.11778G>A or m.3460G>A patients. We did not observe any significant deterioration of VA in patients after treatment discontinuation.

### 3.2. Changes in Visual Field

MD at baseline was similar between groups, but six acute patients had a bilateral decrease in the first year reaching a minimum after 3 to 12 months. At the last available visit MD improved significantly compared to nadir or baseline in acute and chronic groups, respectively (Figure 3). Maximum MD gain during the whole observation period was significant in all groups (acute: +7.3 ± 5.9 dB, *p* = 0.001; early chronic: +2.7 ± 3.4 dB, *p* = 0.003; late chronic: +5.8 ± 4.6 dB, *p* < 0.001). Figure 4 illustrates changes in VF from a late chronic patient during treatment.

### 3.3. Changes in OCT

OCT in acute patients showed a continuous loss of GCL volume and pRNFL thickness during the first year in all eyes (Figure 5). pRNFL loss started at the temporal quadrant and continued to decrease until month 12, reflecting delayed loss of nerve fibers from the other quadrants. GCL volume initially decreased until month 9. From month 12 to last visit there was a small but significant further loss in GCL volume (*p* = 0.048) and pRNFL thickness (*p* = 0.001) in acute patients. Early chronic patients showed a small loss of pRNFL thickness until month 6 (*p* = 0.04) but no change in GCL volume. OCT parameters in late chronic patients remained unchanged during the whole observation period.

### 3.4. Structure–Function Correlations

VA of all patients was not significantly associated with pRNFL thickness or GCL volume at baseline (Figure 6a,b). This was also the case in separate analysis of acute and chronic patients (Appendix A). However, at the last visit both parameters in OCT showed strong correlations with VA (Figure 6c,d). Separate analysis of acute and chronic patients at the last visit also showed significant correlations (Appendix A). VA gain in all patients was weakly associated with GCL volume at baseline (r = 0.36, *p* = 0.01), but not with pRNFL thickness, patient age or disease duration.

MD and CD of all patients at baseline showed weak associations with pRNFL thickness and GCL volume, respectively (Figure 7a,b). However, at the last visit both correlations were much stronger (Figure 7c,d). Separate analysis of acute and chronic patients resulted in similar findings with stronger associations at the last visit (Appendix A).

## 4. Discussion

The present study reflects the experience in treatment of LHON from a single clinical center during the last 5 years. Patients with recent disease onset can have further worsening of VA and VF during the first months of treatment with idebenone. It seems that in some acute patients idebenone is not able to stabilize the progression of RGC dysfunction in the acute phase. On the other hand, some acute patients exhibited a very large improvement in VA after the nadir under continued treatment. This was observed predominantly in patients with the 14484T>C mutation. Since this genotype is known to have a better prognosis of spontaneous visual recovery at a rate up to 58% during the first years [2,19,20], it is not possible to conclusively discriminate between treatment effects and natural course of visual recovery in the acute group. Similarly, the mean increase in VA in early chronic patients could also be attributed to a spontaneous recovery, which is seen in some LHON patients within the first 5 years. However, the CRI observed in both m.11778G>A and three of four m.3460G>A patients of the early chronic group indicates a higher recovery rate under treatment than the previously reported spontaneous recovery rates of 4–25% in the m.11778G>A and 11–40% in the m.3460G>A genotypes [2,19,20,21,22,23]. Finally, a coincidental spontaneous improvement at a rate as observed in six of seven late chronic patients seems even more unlikely. Spontaneous improvement in long-standing LHON has been reported at much lower rates of 9% in m.11778G>A, 0% in m.3460G>A and 19% in 14484T>C genotypes [19]. In addition, the similar increase pattern of VA observed in most chronic patients soon after treatment start must raise a suspicion of causality.

As previous clinical studies of idebenone in LHON examined mostly patients in the acute phase, published consensus only recommended idebenone treatment in the first year after onset [12]. However, recent data from real-world clinical experience showed that nearly half of the recovering patients experience CRI only in the second or third year of treatment, when they have already reached the chronic stage [11]. A post-hoc analysis of the early chronic patients included in the randomized controlled RHODOS trial showed a CRI three times more often in patients treated with idebenone than in the placebo group [24]. We have previously reported a significant improvement in VA in seven of our LHON patients treated in the late chronic phase with an increase of approximately 2 to 3 logMAR lines in at least one eye of each patient during the first treatment year [13]. The current results after extended treatment also show a statistically significant improvement in visual field over time.

Although chronic patients also seem to benefit from idebenone, it has to be emphasized that rapid diagnosis and treatment is of high importance, considering that early treated patients have shown less loss of retinal nerve fibers compared to untreated patients [9], which could enable a better final functional outcome. Nevertheless, our early treated LHON patients featured a substantial loss of RGC and retinal nerve fibers during the first treatment year, which also continued to a lesser degree later on. In chronic patients after more than 5 years from onset, the morphologic parameters in OCT remained unchanged. However, the lack of strong structure-function correlations before treatment, in both acute and chronic patients, indicates a mismatch of visual function and remaining RGC in all stages of LHON. Conversely, there was a strong correlation between inner retinal structure and visual function only after treatment. This was not only seen in acute patients once optic atrophy had fully evolved but also in chronic patients. We interpret this finding as a sign of an optimized signal transduction by reactivated RGC after treatment. It seems likely, that similar reactivation mechanisms may play a role in acute as well as in chronic patients under treatment with idebenone. Since VA gain was only weakly associated with GCL volume before treatment, the amount of inner retinal tissue may not predict the magnitude of improvement in VA that can be achieved by treatment in acute as well as in chronic LHON patients. However, if there is low visual function and less extent of RGC and optic nerve atrophy in a chronic LHON patient, this may indicate that an improvement of visual function can be reached by reactivation of dormant RGC. The functional changes after treatment were not dependent on patient age or disease duration.

The chronic phase of LHON is usually defined as the time when optic atrophy has fully developed and there is no further deterioration of functional and morphological parameters. In most LHON patients, this is the case after the first year. In a minority of patients, a clinically relevant spontaneous improvement can be seen during the early chronic phase and single patients even have full recovery of VA despite substantial loss of RGC and retinal nerve fibers [20,21]. The low number of reported cases of genetically confirmed LHON patients with a substantial spontaneous improvement after more than 5 years from disease onset indicates that such a late spontaneous recovery is a rare phenomenon [19,25,26]. However, since lost optic nerve fibers and RGC cannot regenerate, there is obviously some amount of RGC that survive the acute phase of LHON in a dysfunctional state and can reactivate, sometimes even after a long time. The similar timing of visual improvement in chronic patients after treatment start indicates that idebenone could promote such reactivation.

Recovery of visual function despite optic nerve damage is not unusual after remission of various optic neuropathies and significant improvement following medical treatment has been reported even in chronic optic neuropathies including glaucoma and elapsed anterior ischemic optic neuropathy [27,28]. This illustrates that in acute and even in established chronic optic atrophy certain amounts of RGC are only injured and dysfunctional and have the ability to reactivate signal transduction. Disease specific treatment may therefore be able to restore the function of such injured RGC and hence improve vision.

To date the exact mechanisms how idebenone promotes visual recovery in LHON have not been definitely clarified. Experimental studies using fibroblasts and cybrids carrying LHON mutations have shown that the electrochemical deficits in dysfunctional complex I of the mitochondrial respiratory chain including reduced ATP production and increased amount of ROS can be partly restored by the capacity of idebenone and its metabolites for scavenging ROS and enhancing ATP production [29,30,31]. This mechanism would be important to counteract the propagated oxidative injury that progressively disturbs RGC and optic nerve fibers in the acute stage of LHON [32]. It is also known that the dysfunctional mitochondria of LHON-mutation carriers have an increased mitophagy rate [5] and that the probability of disease conversion increases with reduced mitochondrial count per cell comparing converted to unaffected carriers [33]. Interestingly, idebenone has been shown to reduce mitophagy rate in LHON fibroblasts [5]. This could possibly increase the mitochondrial count per cell and would help to improve OXPHOS in dysfunctional RGC. Finally, histologic studies in optic nerve tissue of LHON patients found amounts of spared axons inside of affected bundles with variable myelination including denuded but viable fibers, and signs indicating remyelination processes [34,35]. Although the effect of idebenone on remyelination has not yet been directly investigated, it showed to stimulate nerve growth factor production in neural tissue [36,37], which is a known pathway of axonal survival and myelin repair [38]. In addition, several other antioxidants have shown enhanced myelin repair and glial activation in models for optic nerve lesions and central neurodegeneration [39,40,41,42]. All of these pathways could contribute to therapeutic effects of idebenone leading to a reactivated signal transduction in surviving dysfunctional ganglion cells in acute as well as in chronic LHON patients (Scheme 1).

The current study has some limitations including its retrospective nature, the small number of patients and unbalanced groups. A sampling bias cannot be excluded and generalization could be limited by the different mutations. However, homogeneous samples in this rare disease are difficult to achieve. The long follow-up period and the regular examination intervals during the observation are strengths of this study. The study was not designed to prove the efficacy of idebenone in LHON treatment, but to explore the impact of treatment on visual function against morphological characteristics for the first time. Since there was no control group available, changes in visual function have to be compared to published natural history data. The increase in VA during treatment exceeded levels of commonly accepted clinical significance in more than half of the examined eyes, which makes possible training or placebo effects rather unlikely.

Since genetic testing has not been widely available until the last 15 years, many patients with long-standing LHON may be genetically undiagnosed and unaware of a treatment option that could promote improvement of visual function even in the late chronic phase. However, larger prospective studies are required to prove the efficacy in these patients.

## 5. Conclusions

Our results support previous evidence that visual function in acute patients and in patients with longer-duration LHON may benefit from treatment with idebenone. The observed changes in chronic LHON patients corroborate the therapeutic potential of idebenone in stabilizing and reconstructing the electrochemical deficits in LHON and in promoting reactivation of dysfunctional RGC injured by this disease.

## Data Availability

The data presented in this study are available on request from the corresponding author.

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
