# Peer review of "Changes in Visual Function and Correlations with Inner Retinal Structure in Acute and Chronic Leber’s Hereditary Optic Neuropathy Patients after Treatment with Idebenone"

_jcm, 2021, doi:10.3390/jcm10010151_

Round 1

Reviewer 1 Report

I would like to thank the authors for dealing with this subject, which remains relevant despite the development of gene therapy in LHON.
Some remarks and points to be corrected.
1 ° Gene therapy should be mentioned in the introduction and recall that idebenone is the only treatment that can be prescribed to all patients carrying LHON regardless of their mutation.
2 ° The OCT protocol must be better detailed. To what degree of eccentricity was the CGG measured? Was the glaucoma or NSite software dedicated to the optic nerve used?
3 ° In the results section, the average evolution is interesting to know.
But it would also be interesting to know if this favorable evolution is the same in all patients. Or are there responder and non-responder patients. If yes how much ?

4 ° It is necessary to consider presenting figures 6 and 7 by separating each of the three subgroups.
5 ° The "previously reported spontaneous recovery rates in those genotypes" must be better defined, by giving quantified values. [

Author Response

Response to Reviewer 1 Comments:

Point 1: Gene therapy should be mentioned in the introduction and recall that idebenone is the only treatment that can be prescribed to all patients carrying LHON regardless of their mutation.

Response 1: We thank the reviewer for this comment and included this information in the revised manuscript:

line 46ff: “Gene therapy for allotopic expression of the wildtype ND4 protein is also currently tested as a treatment in LHON patients, but limited to the most frequent mutation m.11778G>A”

line 51: “…, regardless of the causing mutation“.

Point 2: The OCT protocol must be better detailed. To what degree of eccentricity was the CGG measured? Was the glaucoma or NSite software dedicated to the optic nerve used?

Response 2: We agree with the reviewer and added more information about the OCT measurements including the requested details in the revised manuscript in line 87ff:

“…(Spectralis OCT, NSite module, software version 6.9a, Heidelberg Engineering, Heidelberg, Germany) using a 20×20 degree volume scan centered on the macula, and a 12 degree ring scan centered on the optic nerve head, respectively. Scans at baseline were defined as reference location for all follow-up scans. The built-in segmentation software was used for automated layer segmentation. GCL volume was defined as the layer volume within the central 6 mm diameter of the macula.”

Point 3: In the results section, the average evolution is interesting to know.

But it would also be interesting to know if this favorable evolution is the same in all patients. Or are there responder and non-responder patients. If yes how much ?

Response 3: We agree with this remark and included information about responders and the amount of improvement in responding eyes in Table 1 and in the revised version of the manuscript:

line 127ff: “The percentage of responding eyes with CRI was 56%, 71% and 69% in acute, early chronic and late chronic patients, respectively. VA gain in eyes with CRI was higher in acute patients (Table 1; acute: -0.82 ± 0.39 logMAR from nadir; early chronic: -0.48 ± 0.26 logMAR; late chronic: -0.43 ± 0.29 logMAR; p < 0.037 between acute and chronic groups).”

line142ff: “CRI in at least one eye was reached in all mutations except for the patients with the m.14495A>G or m.13051G>A mutations, who had only mild visual disturbance at treatment start. In the acute group, all eyes of the three patients with the m.14484T>C mutation showed a marked improvement during treatment with normalized VA (≤ 0.0 logMAR) at the last visit in two patients. The other two acute patients with CRI had the mutations m.11778G>A and m.13513G>A, respectively. Among the early chronic cases, both m.11778G>A patients and three m.3460G>A patients showed a CRI. Three early chronic patients reached a final VA ≤ 0.3 logMAR in both eyes. All late chronic patients with the three common LHON mutations had a CRI in at least one eye, but VA increase in patients with the mutation m.14484T>C was not higher than in the m.11778G>A or m.3460G>A patients.”

Point 4: It is necessary to consider presenting figures 6 and 7 by separating each of the three subgroups.

Response 4: We agree with the reviewer and changed Figure 6 and Figure 7 accordingly using different symbols for the three subgroups.

Point 5: The "previously reported spontaneous recovery rates in those genotypes" must be better defined, by giving quantified values.

Response 5: We agree with the reviewer and added the requested information in the text:

line 220: “…recovery at a rate up to 58% during the first years”

line 226f: “…previously reported spontaneous recovery rates of 4-25% in the m.11778G>A and 11-40% in the m.3460G>A genotypes.

line 228ff: “Spontaneous improvement in long-standing LHON has been reported at much lower rates of 9% in m.11778G>A, 0% in m.3460G>A and 19% in 14484T>C genotypes”

Reviewer 2 Report

  • The authors used GCL volume and peripapillary RNFL Thickness for OCT parameters. Is there any reason to use GCL volume instead of GCL thickness or GCIPL thickness or GCC thickness (= GCIPL + NFL thickness)?
  • For the analysis of structure-function orrelation, the authors used Visual field mean deviation versus GCL volume. As the GCL volume is covering the central visual field it would be better to use central visual filed data instead of using global index such as mean deviation.

Author Response

Response to Reviewer 2 Comments:

Point 1: The authors used GCL volume and peripapillary RNFL Thickness for OCT parameters. Is there any reason to use GCL volume instead of GCL thickness or GCIPL thickness or GCC thickness (= GCIPL + NFL thickness)?

Response 1: The analysis of macular ganglion cell layer in the Spectralis software does not provide average thickness for the whole 6 mm measurement area, but calculates single layer volume. However, volume data are equivalent to average thickness since one can be derived from the other using the formula for the volume of a cylinder. GCL measurement more likely represents the amount of ganglion cells in the retina than combined analysis of GCL and RNFL, or GCL and IPL.

Point 2: For the analysis of structure-function correlation, the authors used Visual field mean deviation versus GCL volume. As the GCL volume is covering the central visual field it would be better to use central visual filed data instead of using global index such as mean deviation.

Response 2: We agree with this comment and changed the correlation analyses in Figure 7 and Figure S2 as recommended by the reviewer. Information about the calculation of central deviation is included in the revised manuscript:

line 82ff: “To allow for structure-function analysis in the macula, the six points nasal to the blind spot in the 30 degree test grid were defined as the central cluster of visual field, and central deviation (CD) was calculated as the average from the numerical total deviation map.”

Reviewer 3 Report

I think the authors need to change the research method. 

Because the visual function of LHON can improve even in the natural course, particularly remarkable in mt14484, to research the effectiveness of idebenone, it is essential to compare the medication cases with natural course, not the results of visual function before and after medication. 

(further clarification)

It is insufficient to compare the effects of idebenone, by the method of this study.

The clinical course of LHON varies from the mutation point of mitochondrial DNA. In some cases visual function improve slightly in the natural course. The described results may fall under this category. Even if patients are classified by the disease duration, the effects of idebenone cannot be compared, and the effects of idebeone and retinal structure also cannot be directly compared. When considering the association of idebenone and visual function, the authors need to compare the results of treated and untreated LHON patients first, and verify that the presence of idebenone significantly improves visual function.

Author Response

Response to Reviewer 3 Comments:

Point 1: I think the authors need to change the research method.

Because the visual function of LHON can improve even in the natural course, particularly remarkable in mt14484, to research the effectiveness of idebenone, it is essential to compare the medication cases with natural course, not the results of visual function before and after medication.

Response 1: We thank the reviewer for this critical comment and agree, that visual function can improve in the natural course of LHON. This is a relevant problem in all studies investigating LHON treatment in the early phase. Our study was not designed to prove the efficacy of idebenone. We aim to explore associations of neuroretinal structure and visual function before and after treatment, because no previous information is available about these correlations. For this purpose, information about significant changes in visual function and retinal measurements must also be included. We agree with the reviewer that changes in treated patients should be better compared to the natural course and included more information about improvement rates in untreated patients from previous reports:

line 220: “this genotype is known to have a better prognosis of spontaneous visual recovery at a rate up to 58% during the first years“

line 224ff: “However, the CRI observed in both m.11778G>A and three of four m.3460G>A patients of the early chronic group indicates a higher recovery rate under treatment than the previously reported spontaneous recovery rates of 4-25% in the m.11778G>A and 11-40% in the m.3460G>A genotypes.”

line 228ff: “Spontaneous improvement in long-standing LHON has been reported at much lower rates of 9% in m.11778G>A, 0% in m.3460G>A and 19% in 14484T>C genotypes.”

We clarified that treatment effects in the acute patients cannot be defined, because of included m.14484 patients:

line 221f: “Since this genotype is known to have a better prognosis of spontaneous visual recovery at a rate up to 58% during the first years [2,19,20], it is not possible to conclusively discriminate between treatment effects and natural course of visual recovery in the acute group.”

However, the improvement rates in chronic patients during treatment were higher than in all available reports of the natural course of LHON (as stated in line 224ff).

(further clarification)

Point 2: It is insufficient to compare the effects of idebenone, by the method of this study.

The clinical course of LHON varies from the mutation point of mitochondrial DNA. In some cases visual function improve slightly in the natural course. The described results may fall under this category. Even if patients are classified by the disease duration, the effects of idebenone cannot be compared, and the effects of idebeone and retinal structure also cannot be directly compared. When considering the association of idebenone and visual function, the authors need to compare the results of treated and untreated LHON patients first, and verify that the presence of idebenone significantly improves visual function.

Response 2: We thank the reviewer for the chance to comment on this objection. We agree that the natural course of LHON varies in different mutations and included information concerning this matter as stated in Response 1.

We also agree that visual function can improve in the natural course in a minority of cases. However, we did not observe slight, but clinically relevant and statistically significant changes during treatment and this happened at a much higher rate than previously reported spontaneous improvement in chronic LHON (see Response 1 above). In addition, these changes coincided with treatment in the improving patients, which indicates a probable causality:

line 230f: “In addition, the similar increase pattern in most chronic patients soon after treatment start must raise a suspicion of causality.”

line 271f: “The similar timing of visual improvement in chronic patients after treatment start indicates that idebenone could promote such reactivation.”

Most previous studies with idebenone classified LHON patients by the disease duration: The retrospective study of Carelli V. et al. (Brain. 2011;134:e188, doi:10.1093/brain/awr180) included only patients within 1 year after onset, to avoid the period in which the probability of spontaneous recovery of visual acuity is higher (2 to 5 years from onset). The RHODOS trial (Klopstock T. et al. Brain 2011;134:2677–86, doi:10.1093/brain/awr170) examined the effects of idebenone on retinal structure in subgroups classified by disease duration and data from this trial were recently analyzed to separately investigate treatment response in chronic LHON patients:

line 236: “A post-hoc analysis of the early chronic patients included in the randomized controlled RHODOS trial showed a CRI three times more often in patients treated with idebenone than in the placebo group.”

A recently published prospective study also compared early intervention patients (within 1 year of onset) and late intervention patients (after 1 year of onset) during idebenone treatment and reported different results in acute and chronic groups (Ishikawa H. et al. Jpn J Ophthalmol. 2020 Nov 13. doi: 10.1007/s10384-020-00789-2).

The reason for classification of LHON patients was added in the revised manuscript:

line 94f: “Since visual function in LHON is known to develop differently during the successive stages, patients were divided into three groups according to their time from LHON onset”

As mentioned in Response 1, this trial was not designed to prove the efficacy of idebenone in visual loss due to LHON. A significant effect of idebenone in treated LHON patients compared to untreated patients has already been shown in the placebo-controlled RHODOS trial. We agree that controlled studies would be generally preferable. However, since idebenone is an approved treatment for LHON in many countries including our own, such a trial would be ethically problematic. In addition, it seems not feasible to include patients stratified by mutations and disease stages in a controlled study.

We revised the limitation paragraph addressing the reviewer’s comments:

line 304ff: “The current study has some limitations including its retrospective nature, the small number of patients and unbalanced groups. A sampling bias cannot be excluded and generalization could be limited by the different mutations. However, homogeneous samples in this rare disease are difficult to achieve.”…”The study was not designed to prove the efficacy of idebenone in LHON treatment, but to explore the impact of treatment on visual function against morphological characteristics for the first time. Since there was no control group available, changes in visual function have to be compared to published natural history data.”

Round 2

Reviewer 1 Report

I would like to thank the authors of this article for their responses.

The article can be accept as it is 

Reviewer 3 Report

Considering the rarity of LHON, it is appropriate to publish this paper as one of the research results.